# *Serendipita indica* Promotes the Growth of Tartary Buckwheat by Stimulating Hormone Synthesis, Metabolite Production, and Increasing Systemic Resistance

**DOI:** 10.3390/jof9111114

**Published:** 2023-11-17

**Authors:** Meijia Zheng, Shanpu Zhong, Wenjing Wang, Zizhong Tang, Tongliang Bu, Qingfeng Li

**Affiliations:** College of Life Sciences, Sichuan Agricultural University, Ya’an 625014, China; meijiazheng3426@163.com (M.Z.); heavenmile@163.com (S.Z.); wangwj322023@163.com (W.W.); 14126@sicau.edu.cn (Z.T.); tlbu@sicau.edu.cn (T.B.)

**Keywords:** Tartary buckwheat, *Serendipita indica*, growth performance

## Abstract

The main objective of this study was to investigate the influence of *Serendipita indica* on the growth of Tartary buckwheat plants. This study highlighted that the roots of Tartary buckwheat can be colonized by *S. indica* and that this fungal endophyte improved plants height, fresh weight, dry weight, and grain yield. In the meantime, the colonization of *S. indica* in Tartary buckwheat leaves resulted in elevated levels of photosynthesis, plant hormone content, antioxidant enzyme activity, proline content, chlorophyll content, soluble sugars, and protein content. Additionally, the introduction of *S. indica* to Tartary buckwheat roots led to a substantial rise in the levels of flavonoids and phenols found in the leaves and seeds of Tartary buckwheat. In addition, *S. indica* colonization reduced the content of malondialdehyde and hydrogen peroxide when compared to non-colonized plants. Importantly, the drought tolerance of Tartary buckwheat plants is increased, which benefits from physiology and bio-chemical changes in plants after *S. indica* colonized. In conclusion, we have shown that *S. indica* can improve systematic resistance and promote the growth of Tartary buckwheat by enhancing the photosynthetic capacity of Tartary buckwheat, inducing the production of IAA, increasing the content of secondary metabolites such as total phenols and total flavonoids, and improving the antioxidant enzyme activity of the plant.

## 1. Introduction

Tartary buckwheat (*Fagopyrum tataricum* Gaertn.) is a group of pseudocereals of the genus *Fagopyrum* in the *Polygonaceae* family. Tartary buckwheat is considered an important medicinal and edible food crop. It is rich in proteins and flavonoids, has hypoglycemic and anti-inflammatory effects, and can enhance the immunity of the body, which is quite beneficial to one’s health [1,2,3]. A survey of the Tartary buckwheat cultivation in China shows that it can grow in high-altitude areas in the southwest [4]. Sichuan Yi Autonomous Prefecture in China is located in the center of cultivated Tartary buckwheat [5]. As one of the main crops, the growth and development of Tartary buckwheat face issues such as poor planting areas, low soil fertility, drought, and salt stress in the plantation areas, resulting in low emergence rates and poor seedling quality, ultimately affecting the production of Tartary buckwheat. Therefore, it is necessary to find an efficient, environmentally friendly, and promotional way to solve the growth problem of Tartary buckwheat. Many studies have highlighted that several endophytic fungal microorganisms improve their plant hosts’ vegetative performance and their resistance to biotic and abiotic stresses. Endophytic fungal microorganisms have also recently been identified in Tartary buckwheat, with beneficial effects in the same plants. Researchers have examined the influence of soil microorganisms on endophytic fungi in Tartary buckwheat and determined that firmicutes is the predominant microbial group of Tartary buckwheat, which is passed down through generations [6]. A previous study has indicated that the endophytic fungus *Fusarium oxysporum* Fat9 enhances the growth, accumulation of flavonoids, and antioxidant properties of Tartary buckwheat through the use of water-extracted mycelia polysaccharide (WPS), sodium hydroxide-extracted mycelia polysaccharide (SPS), hydrochloric-extracted mycelia polysaccharide (APS), and exo-polysaccharide (EPS) [7]. Meanwhile, studies have shown that endophytic fungi *Fusarium oxysporum* Fat9 and *Alternaria* sp. Fat15 can stimulate the phenylpropanoid pathway and promote an increase in Tartary buckwheat rutin content [8]. Thus, endophytic fungi can promote the growth of Tartary buckwheat and accumulate its metabolites.

As an endophytic fungus in plant roots, *Serendipita indica* has a wide range of hosts and can interact with a variety of plants to promote their growth and enhance stress tolerance [9]. It belongs to Basidiomycota, Agaricomycotina, Hymenomycetes, Sebacinales, Sebacinaceae and *Serendipita*. In recent years, the endophytic fungus *S. indica* in plant roots has received a great deal of attention due to its wide colonization range and promoting effects on plants. According to existing data, the number of plants colonized by *S. indica* exceeds 200, and it has been clearly shown to have promoting effect on 24 host plants in 12 families [10,11]. The fungus is easily cultivable, lacks host specificity, and colonizes the roots of many different plants, for the most part in an endophytic manner [12]. *S. indica* is a kind of biological nutrient symbiotic root system that can promote plant growth [13]. It was originally isolated from the roots of a shrub in the Thar desert in northwest India by Indian scientist Verma in 1998. Studies showed that it had the ability to simulate typical arbuscular mycorrhizal fungi (AMF) [12,14]. Colonization by *S. indica* has been reported to help in increasing crop yield [15,16,17], promote host plant growth, improve crop quality [18,19,20], stimulate the accumulation of plant metabolites [21], and enhance resistance to host plant stresses [22,23,24,25,26]. Since the discovery of *S. indica*, research into their growth-promoting effects on plants and on their ability to enhance the stress resistance of host plants has been continuously developing.

The objective of this study was to determine the influence of *S. indica* on the growth of Tartary buckwheat plants. The growth parameters and some physiological parameters were measured in *S. indica*-colonized Tartary buckwheat plants. The physiological and biochemical changes of colonized and non-colonized Tartary buckwheat of *S. indica* were also observed under drought conditions. In conclusion, the results of this study contribute to the elucidating the growth promoting and stress resistance mechanism of *S. indica* in Tartary buckwheat, and provide a foundation for the future application of *S. indica* in the production of Tartary buckwheat.

## 2. Materials and Methods

### 2.1. Plant Materials and Fungi Preparation

The seed liquid selection method (a method that uses clean water or various solutions with different relative densities to select seeds) is used to screen the seeds of Tartary buckwheat, remove the dried seeds, and soak the selected plump seeds in 42 °C water for 2 h before use. *S. indica* was propagated on solid CM medium and incubated at 28 °C for 2 weeks. For inoculation, 200 mL of CM liquid medium was supplied by *S. indica* with a diameter of 7.5 mm and incubated for 20 days at 28 °C and 120 rpm on a rotary shaker. For plant inoculation, pure white mycelium was collected and washed 3 times with sterile double distilled water, 2 g of *S. indica* mycelium was mixed with 100 mL sterilized double distilled water to produce a 2% (*w*/*v*) *S. indica* suspension (±1.5 × 10^7^ spores/mL) [27]. One group was treated with freshly prepared *S. indica* chlamydospore suspension at a final content of 100 mL/kg soil, and the other group was treated with distilled water and used as controls [19,28]. The *S. indica* group and the control group have eight pots, with four plants per pot.

### 2.2. Microscopy and Detection of S. indica in Colonized Roots

The mixed soil was sterilized at 120 °C for 20 min with an autoclave. Soak Tartary buckwheat seeds in water, remove seed skins, disinfect them with 0.1% mercuric chloride solution for 5–10 min, rinse with sterile water after removal and set aside. After 15 days of co-cultivation, roots of *S. indica* treated Tartary buckwheat were collected and washed under running water to remove the attached nutrient soil, cut into 1 cm segments, soaked in 10% KOH at room temperature for 30 min, soaked in 1% HCl solution for 20 min, stained with 0.02% Trypan Blue solution for 2 h, and finally decolorized with 50% lactophenol solution for 2 h [29]. Then, roots were washed with sterile water three times. After removing the surface water with filter paper, the *S. indica* colonization in root segments was observed under an Olympus fluorescence microscope BX53 (Olympus, Shanghai, China).

### 2.3. Plant Growth-Related Parameters

After about 40 days of co-cultivation, 4 plants were taken from each pot to measure or calculate growth related parameters, such as plant height, leaf width, leaf length, petiole length, fresh and dry weight of aboveground parts, and root fresh and dry weight of both colonized and non-colonized control seedlings. Measure the height of the Tartary buckwheat plant with a line, and then measure the length of the line with a ruler to obtain the height of the plant. The following methods were used to measure leaf width, leaf length, and petiole length. Leaf width was measured at the widest part of the Tartary buckwheat leaf. The distance from tip to petiole was the length of the leaf. The fresh weight of Tartary buckwheat overground parts and roots were weighed using an MTQ300 electronic scale (Meilun, Shenzhen, China). The plant samples were dried to constant weight in an oven at 80 °C and weighed using an MTQ300 electronic scale (Meilun, Shenzhen, China) to obtain the dry weight of the plants.

### 2.4. Determination of Chlorophyll and Photosynthesis Parameters

The contents of chlorophyll A, chlorophyll B, total chlorophyll in leaves of *S. indica*-colonized and non-colonized control seedlings were determined. Clean fresh plant leaves and absorb surface moisture. Grind them into powder with liquid nitrogen under dark conditions. Soak 0.1 g of plant leaf samples in 5 mL of acetone–ethanol mixed extracting solution (acetone/ethanol = 2:1, *v*/*v*) and in dark for more than 3 h until the leaves completely turn white. Then, centrifuge the supernatant and use the blank extract solution as the control. With absorbance at 663, 645 nm wavelengths was determined using a V-1100D spectrophotometer (MAPADA, Jinan, China). The concentrations of various pigments were calculated from the absorbance values using the same equations described by Arnon [30].

The fluorescence imaging system GFS-3000 (WALZ, Nuremberg, Germany) was used to measure the transpiration rate (Tr), stomatal conductance (Gs), intercellular CO_2_ concentration (Ci), and net photosynthetic rate (Pn) of plants colonized and non-colonized by *S. indica*. Three plants were tested for each treatment, and one well-grown leaf from the same location was selected for each plant.

### 2.5. Determination of Total Phenols and Flavonoids Content

To determine the total phenolic content of both *S. indica*-colonized and non-colonized Tartary buckwheat leaves, 0.1 g of leaf sample was taken, and the leaves were extracted with 2 mL of 60% ethanol. After vortex mixing, they were extracted at 60 °C for 30 min and centrifuged at 4000 rpm for 10 min to collect the supernatant. Then, the total phenolic content of Tartary buckwheat leaves was determined using a plant total phenolic test kit (colorimetric method) (Nanjing Jiancheng Bioengineering Institute, Nanjing, China) and an enzyme-labeled instrument, Multiskan SkyHigh 500C (ThermoFisher Scientific, Shanghai, China). For each parameter, three replications were made.

Firstly, extract total flavonoids from Tartary buckwheat leaves and seeds. Weigh 0.1 g of sample powder with dry weight, add 5 mL of 70% ethanol at 40 °C to a water bath for ultrasonic treatment for 2 h, centrifuge at 12,000 rpm for 5 min to collect the supernatant, add Petroleum ether for fractionation, collect the lower liquid, repeatedly extract until the upper Petroleum ether is clarified, and pass through a 0.45 nm filter membrane to obtain total flavonoids. Then, the total flavonoid content was determined using the AlCl_3_ colorimetric method [31]. Take 0.1 mL of flavonoid sample, add 2 mL of AlCl_3_ solution, mix well, and let stand for 10 min. Add 3 mL of KOAc solution and dilute to 10 mL with 70% ethanol. After mixing, let stand at room temperature for 30 min for testing. Measure the absorbance of each sample at a wavelength of 420 nm. Compare the standard curve of rutin and calculate the total flavonoid content.

### 2.6. Drought Treatment and Stomatal Response Analysis of Tartary Buckwheat

After the co-cultivation of *S. indica* and Tartary buckwheat for 30 days, both colonized and non-colonized Tartary buckwheat plants were subjected to natural drought treatment until their leaves wilted. After treatment, it was divided into four groups: (1) non-drought treatment (control) and non-colonized (CK) groups; (2) non-drought treatment (control) and colonized (P+) groups; (3) drought treatment (drought) and non-colonized (CK) groups; (4) drought treatment (drought) and colonized (P+) groups. (Stomatal density (number/m^2^) = number of stoma/image area).

Apply transparent nail polish on the leaf epidermis of Tartary buckwheat at the same position, and then place the nail polish transfer film on the slide. Observe the stomatal density and opening/closing of leaves under a microscope.

### 2.7. Detection of IAA Content in S. indica

The potency of culturable endophytic fungi as an IAA producer was determined according to the method described by [32] with minor modifications. Inoculate *S. indica* into king B culture medium containing and without tryptophan for seven days. Take 2 mL of bacterial solution at 12,000 rpm and centrifuge for 15 min. Take 1 mL of supernatant and add an equal amount of PC colorimetric solution. React in the dark for 30 min and observe color changes.

### 2.8. Determination of Antioxidant Enzyme Activity and Antioxidant-Related Molecular Content

To determine the antioxidant enzyme activity and antioxidant-related molecules content of both colonized and non-colonized Tartary buckwheat leaves, 0.1 g of fresh plant leaves were taken, and 9 times the volume of physiological saline was added. 10% tissue homogenate was prepared under ice bath conditions and centrifuged at 3500 rpm for 10 min, and the supernatant was taken for testing. The activity of antioxidant enzymes SOD, CAT, GSH-Px, and POD in Tartary buckwheat leaves, as well as the content of MDA, Pro, H_2_O_2_, GSH, SS, and SP, were measured using the relevant reagent kit from Nanjing Jiancheng Bioengineering Institute (Nanjing, China). For each parameter, three replications were made.

### 2.9. RNA Isolation and Quantitative RT-PCR Analysis

Total RNA was extracted from 50 mg of frozen plant samples using a FastPure Universal Plant Total RNA Isolation Kit (Vazyme, Nanjing, China) according to the manufacturer’s protocol. RNA was used to synthesize complementary (c) DNA using a PrimeScript™ RT reagent Kit with gDNA Eraser (Takara Bio Inc., Shiga, Japan) based on the manufacturer’s instructions. With TB Green ™ Premix Ex Taq ™ II (Tli RNaseH Plus) (Takara Bio Inc., Shiga, Japan), use gene specific primers to amplify genes according to the manufacturer’s instructions.

The real-time quantitative (qRT)-PCR amplification program consisted of 1 cycle at 95 °C for 3 min, followed by 39 cycles of 95 °C for 10 s, and 56 °C for 30 s, and the gene expressions were calculated by 2^−ΔΔct^ method. To normalize the gene transcript level, the Tartary buckwheat *FtCACS* gene was co-amplified as a reference gene. The gene-specific primers are shown in Table 1.

### 2.10. Analysis of Plant Hormone Metabolism Pathway

Take 1 g of fresh Tartary buckwheat roots, stems, and leaves, both colonized and non-colonized by *S. indica*, wash them with double distributed water, dry them, centrifuge them in a suitable size, freeze them in liquid nitrogen for 5–10 min, and store them in a refrigerator at −80 °C. The preserved samples were sent to METWARE Biotechnology Company (Wuhan, China) for plant hormone related metabolic analysis using LC-MS/MS. Plant hormone correlation analysis mainly measured the content of IAA, ACC, ABA, GA, JA, and SA hormones in the roots, stems, and leaves of *S. indica*-colonized and non-colonized buckwheat plants.

### 2.11. Statistical Analyses

All graphs were created, and statistical calculations were performed using GraphPad Prism 8.0. The significance of the data obtained was checked by Student’s *t*-test using the program IBM SPSS Statistics 26. Data are means of at least three independent biological repeats with standard errors.

## 3. Results

### 3.1. Detection of S. indica Colonization and Growth Changes of Tartary Buckwheat

*S. indica* colonization in Tartary buckwheat roots was detected by observing root sections under a microscope after trypan blue staining. After 15 days of co-cultivation of Tartary buckwheat and *S. indica*, the spores and mycelium of *S. indica* can be observed in the roots of Tartary buckwheat (Figure 1). This indicates that *S. indica* can colonize Tartary buckwheat plants, and that the *S. indica*-colonized seedlings were used in further experiments.

After 40 days of co-cultivation, the phenotypes of *S. indica*-colonized and non-colonized Tartary buckwheat plants were measured. The colonization of *S. indica* not only significantly increased the aboveground fresh weight of Tartary buckwheat (*p* < 0.01), but also extremely significantly improved other growth indicators (*p* < 0.001) (Figure 2, Table 2). In addition, weighing the mature seeds of *S. indica*-colonized and non-colonized Tartary buckwheat revealed a significant increase in the hundred-grain weight of *S. indica*-colonized Tartary buckwheat seeds (*p* < 0.01) (Table 2). Colonization by *S. indica* promotes the overall growth of Tartary buckwheat plants.

### 3.2. Effects of S. indica on Photosynthetic Pigments and Cellular Respiration of Tartary Buckwheat

Photosynthesis is the main method of plant organic matter accumulation. Chlorophyll plays an important role in photosynthesis, such as providing oxygen and converting light energy. Therefore, we measured the chlorophyll content and cellular respiration related parameters of Tartary buckwheat plants colonized and non-colonized by *S. indica*. The total chlorophyll content, chlorophyll A, and chlorophyll B of Tartary buckwheat colonized seedlings by *S. indica* were extremely significantly higher than the control group (*p* < 0.001), with increases of 56.14%, 37.23%, and 51.15%, respectively (Figure 3A). The photosynthetic pigments content in leaves of the two groups was significantly different, and the related parameters of cellular respiration were further determined for the Tartary buckwheat leaves.

Plants undergo photosynthesis under light and absorb CO_2_ through their stomata, so the stomata must open. However, the opening of stomata inevitably leads to transpiration, and the stomata can adjust the size of the opening according to changes in environmental conditions, allowing plants to obtain the most CO_2_ under less water-loss conditions. *S. indica* colonization significantly increased transpiration rate (Tr) (Figure 3B) and stomatal conductance (Gs) (Figure 3C) (*p* < 0.01), increasing by 122.68% and 133.45%, respectively. There was a difference in net photosynthetic rate (Pn) (Figure 3D) (*p* < 0.05), but it was not significant, increasing by 68.85%. The intercellular CO_2_ concentration (Ci) slightly increased to 4.61% (Figure 3E).

### 3.3. Effect of S. indica on Plant Hormone Content of Tartary Buckwheat

In order to understand the growth promoting mechanism of *S. indica* colonization on the root system of Tartary buckwheat, we measured the content of different plant hormone in the root, stem and leaf of Tartary buckwheat. The results showed that after the colonization of *S. indica*, the IAA in the roots, stems, and leaves of Tartary buckwheat plants significantly increased, which were 10.6-fold, 1.7-fold, and 1.5-fold higher than the control group, respectively (Figure 4). The evaluation of the growth promoting function of *S. indica* revealed that they can produce IAA. ACC is an important precursor for ethylene synthesis. The ACC content in the roots, stems, and leaves of *S. indica* colonization was 2.7-fold, 2.3-fold, and 1.1-fold higher than that of the control group (Figure 4). IAA and ACC have a synergistic effect and can jointly regulate plant growth and development [33]. The colonization of *S. indica* significantly increased the IAA and ACC content of Tartary buckwheat, promoting their growth. Meanwhile, in the roots of Tartary buckwheat colonized by *S. indica*, the content of ABA and JA increased, while the content of GA and SA decreased. In the stem, the content of ABA and SA increased, while the content of GA and JA decreased, and in the leaves, the content of ABA, JA, and SA increased, while the content of GA decreased (Figure 4).

### 3.4. Effect of S. indica on the Synthesis of Auxin in Tartary Buckwheat

IAA is the main form of plant auxin and plays multiple roles in plant growth and development. It mediates the elongation of stem and root growth, the enlargement of fruits and tubers, and the promotion of cell division, through regulating cell division, expansion, differentiation, and patterning [34]. IAA is the best-studied naturally occurring active auxin. IAA biosynthesis can occur through two major routes: tryptophan (Trp)-dependent and Trp-independent pathways [35]. The classification and enrichment analysis results of KEGG pathway types with significantly different metabolites showed that the colonization of Tartary buckwheat by *S. indica* affected tryptophan metabolism, Plant hormone signal transduction and other pathways (Figure 5A). Among Trp-dependent pathways, the IPyA pathway is the main contributor to IAA and the only pathway in which every step from Trp to IAA has been identified [36].

*S. indica* themselves can produce IAA. And after the being colonized by *S. indica*, the IAA content in the roots, stems, and leaves of Tartary buckwheat also showed an upward trend. In order to understand the reason for the increase in IAA content in Tartary buckwheat after the colonization of *S. indica*, we measured the expression level of genes related to the IAA synthesis pathway in Tartary buckwheat leaves. We quantitatively analyzed the *NIT*, *AMI1*, and *YUCCA* genes in the IPyA pathway of leaves of *S. indica*-colonized and non-colonized Tartary buckwheat. At the same time, we also quantitatively analyzed the *ARF2* gene that controls leaf senescence and fruit development [37]. The results showed that the colonization of Tartary buckwheat with *S. indica* significantly increased the expression levels of *NIT*, *AMI1*, *YUCCA*, and *ARF2* genes (*p* < 0.01) ((Figure 5B).

### 3.5. Effects of S. indica on Metabolites of Tartary Buckwheat

The KEGG enrichment analysis of significant difference metabolites showed that the colonization of *S. indica* affected the synthesis of secondary metabolites and other metabolic pathways of Tartary buckwheat (Figure 5A). So, we measured the soluble sugar (Figure 6A) and soluble protein (Figure 6B) content of *S. indica*-colonized and non-colonized Tartary buckwheat. The colonization of Tartary buckwheat by *S. indica* significantly increased the soluble sugar content in the leaves of Tartary buckwheat (*p* < 0.01); the soluble protein content also increased (*p* < 0.05). At the same time, due to the rich content of flavonoids (Figure 6D) and phenolic (Figure 6C) substances in Tartary buckwheat [38], we also measured the total flavonoids and total phenols content of the leaves and seeds of *S. indica*-colonized and non-colonized Tartary buckwheat. The results showed that the total flavonoid and total phenolic contents in the leaves and seeds of Tartary buckwheat extremely significantly increased after colonization by *S. indica* (*p* < 0.001).

### 3.6. Effects of S. indica Colonization on the Antioxidant System of Tartary Buckwheat under Drought Stress

Flavonoids are synthesized and accumulate in a variety of tissues in many plant species and can help to protect against various stresses [39]. The colonization of Tartary buckwheat by *S. indica* significantly increased the total flavonoid content in its leaves. Therefore, we subjected *S. indica*-colonized and non-colonized Tartary buckwheat plants to drought stress. Under natural drought conditions, the overall phenotype of *S. indica*-colonized Tartary buckwheat is better than that of non-colonized plants (Figure 7A). 

Observing the number and opening/closing of stomata under normal growth and drought conditions, the results showed that within the same field of view, the air density and opening and closing of Tartary buckwheat leaves were different (Figure 7). The results showed that the *S. indica* significantly increased the stomatal densities of Tartary buckwheat leaves under both normal and drought conditions (Figure 7F). When facing drought, plants reduce water transpiration by adjusting the stomatal opening/closing state on the surface of their leaves. The *S. indica* colonization of Tartary buckwheat increased the number of stomata per unit area of Tartary buckwheat leaves, allowing Tartary buckwheat to reduce water loss and cope with drought stress by regulating stomata.

We further measured the activity of a series of antioxidant enzymes and the content of antioxidant related molecules in the leaves of *S. indica*-colonized and non-colonized Tartary buckwheat under drought stress. At the same time, quantitative analysis was conducted on genes related to antioxidant activity. Regardless of the conditions, the colonization of *S. indica* significantly increased the GSH content (Figure 8A) and decreased the H_2_O_2_ content in Tartary buckwheat leaves (Figure 8B) (*p* < 0.001). The MDA content of *S. indica*-colonized Tartary buckwheat extremely significantly decreased under drought conditions (*p* < 0.001), and even under non drought conditions, the MDA content in the leaves also decreased (*p* < 0.05) (Figure 8C). The content of plant osmoregulation factor Pro extremely significantly increased under drought conditions (*p* < 0.001). Under normal conditions without drought, the colonization of *S. indica* also led to an increase in Pro content in Tartary buckwheat plants (*p* < 0.05) (Figure 8D). 

Under drought conditions, the colonization of Tartary buckwheat by *S. indica* extremely significantly increased the activities of antioxidant enzymes GSH-Px, CAT, SOD, and POD in the leaves of Tartary buckwheat (*p* < 0.001). Under normal conditions without drought, GSH-Px activity extremely significantly decreased (*p* < 0.001), CAT activity increased (*p* < 0.05), SOD activity extremely significantly increased (*p* < 0.001), and POD activity significantly increased (*p* < 0.01) (Figure 9A–D). The results of gene quantitative analysis showed that the colonization of Tartary buckwheat by *S. indica* increased the expression levels of *FtSOD1* and *FtPOD1* genes in the leaves of Tartary buckwheat (*p* < 0.05); The expression levels of *FtSOD2* and *FtP5CS* genes significantly increased (*p* < 0.01); The expression level of the *FtCAT* gene extremely significantly increased (*p* < 0.001) (Figure 9E).

## 4. Discussion

Crops are colonized by complex microbial communities, and some of them are detrimental and cause diseases, whereas others promote plant growth and enhance nutrient acquisition as well as tolerance to biotic and abiotic stresses via a multitude of mechanisms [40]. Root-symbiotic microbes can improve nutrient uptake, plant biomass and yield, and may enhance tolerance of host plants to various environmental stresses [40,41]. Tartary buckwheat, as one of the main food crops, encounters a series of problems, such as poor soil quality, during its cultivation. As an efficient and harmless endophytic fungus, *S. indica* can be effectively used to solve the problems encountered in the cultivation of Tartary buckwheat.

It was proved that the plant-growth promoting effect of *S. indica* was achieved by increasing the length and number of roots [18,42]). Previous research has also shown that colonization of *S. indica* can increase the root length, root area, and root weight of plants such as strawberries [15], bananas [28], and longans [20]. While favorable effects of the root-endophytic fungus *S. indica* on a wide range of plant species has been reported, we demonstrated its potential plant-growth promoting effect on Tartary buckwheat. The results of this study showed that *S. indica*-colonized the roots of Tartary buckwheat, increasing the biomass of Tartary buckwheat roots (Table 2). This result indicates that *S. indica* colonization has a promoting effect on the growth of Tartary buckwheat plant roots. The roots have functions such as uptake, support, and storage. *S. indica* indirectly promoted the growth of Tartary buckwheat plants by promoting the growth of roots, and also have a strong promoting effect on the aboveground parts of Tartary buckwheat (Figure 2, Table 2).

The leaves of plants are the specialized organs for capturing light energy, which is used to support photosynthesis. During this process, the carbon dioxide in the atmosphere is incorporated into the organic compounds of plants, which in turn promotes plant growth [43]. Research shows that *S. indica* promotes the leaf enlargement of Tartary buckwheat (Figure 2), thus promoting plant photosynthesis. *S. indica* can promote the increase in photosynthetic pigment content in its host plants, enhance photosynthesis, and ultimately enhance plant growth [15,16]. We further measured the photosynthetic pigment content of *S. indica*-colonized and non-colonized Tartary buckwheat and found that *S. indica*-colonized Tartary buckwheat extremely significantly increased the chlorophyll A, chlorophyll B, and total chlorophyll content of Tartary buckwheat plants (Figure 3A). These promoting effects on chlorophyll may be due to the enhanced nutrient uptake and transportation of *S. indica* colonization roots. In parallel, we also measured the relevant parameters of cellular respiration, and found that the colonization of *S. indica* also improved the photosynthetic capacity of Tartary buckwheat (Figure 3). Elevated cellular respiration can increase the accumulation of dry matter in Tartary buckwheat, thereby promoting the growth of Tartary buckwheat.

Endogenous plant hormones play an important role in regulating plant growth and root development. IAA is a key regulator of growth, playing multiple roles in almost every aspect of plant growth and development, and is linked to the establishment of a symbiotic relationship between the endophyte and the plant [44]. *S. indica* has been shown to promote plant growth by stimulating the synthesis of host plant IAA [45,46]. However, it has been also pointed out that the endophytic fungus *S. indica* in plant roots has different effects on the synthesis of IAA and related genes expression in different host plants [47]. Therefore, we assessed the change in plant hormone content in Tartary buckwheat colonized by *S. indica*. Before this, we found that *S. indica* have the ability to produce IAA (Figure 4A). It was found that IAA levels were increased in the roots, stems, and leaves of Tartary buckwheat plants colonized by *S. indica* (Figure 4B). Ethylene precursor substances, ACC and IAA, can interact synergistically or antagonistically, controlling various plant developmental processes, including root formation and hypocotyl elongation [33]. The colonization of Tartary buckwheat by *S. indica* resulted in an increase in both the content of IAA and ACC in its root, which may also be used to explain the increase in Tartary buckwheat’s root system. In addition, we tested the expression levels of the IAA-synthesis-related genes. Here, significant increases were observed in the expression of *NIT*, *AMI1*, and *YUCCA* (Figure 5B). While simultaneously measuring the *ARF2* gene related to plant growth and fruit development, the expression level of ARF2 was found to be significantly increased as well (Figure 5B). Our results showed that the colonization of *S. indica* in Tartary buckwheat promoted the growth of Tartary buckwheat plants by stimulating the biosynthesis of IAA as well as the expression of genes related to synthesis pathway. 

Plant hormones have also been shown to have a regulatory effect on flavonoid biosynthesis, the content of flavonoid in *Arabidopsis* roots is increased by IAA as well as by ACC [48]. This study found that change fold of IAA and ACC were increased in the roots, indicating that the colonization of Tartary buckwheat by *S. indica* resulted to an increase in the content of IAA and ACC. Indeed, the colonization of *S. indica* promoted the content of total flavonoids and total phenols in the leaves and seeds of Tartary buckwheat (Figure 6), respectively. Flavonoids such as anthocyanidin are considered as active oxygen scavengers, which play an anti-inflammatory role in plant biologic and abiotic stress [49]. The biosynthesis of anthocyanin in *Arabidopsis thaliana* is affected by other growth regulators, such as ABA (abscisic acid), GA3 (gibberellic acid), and JA (jasmonic acid) [50]. GA has been shown to inhibit the biosynthesis of flavonoids in *Arabidopsis thaliana* [51]. JA has an inducing effect on anthocyanin synthesis in various plants [52]. ABA is known to be involved in numerous stress-adaptation processes, including the promotion of anthocyanin biosynthesis [53]. Studies have found that the level of JA in host roots responded differently to root colonization by *S. indica* [54]. In this study, the change fold of ABA and JA due to the *S. indica* colonization of Tartary buckwheat significantly increased, indicating that *S. indica* colonization promoted the synthesis of flavonoids such as anthocyanins in Tartary buckwheat roots (Figure 4). It is impossible that the accumulation of higher flavonoid in leaves may help Tartary buckwheat to withstand stress, thus promoting the growth of Tartary buckwheat.

Our research results show that colonization of *S. indica* can increase the chlorophyll content and promote photosynthesis in Tartary buckwheat leaves (Figure 3). Photosynthesis is the main way of material accumulation [43]. Soluble sugars are the basis of plant metabolism and an important energy source during plant growth and development. They can provide energy and metabolic intermediate for plant growth and development, promote the germination of plant seed, and promote early seedling development. As well as being an important regulator of plant growth, development and gene expression, as a small molecular organic compound, soluble sugar is major solute involved in regulating the osmotic pressure in plant cells. Plants subjected to osmotic stress will actively accumulate solutes in order to reduce osmotic potential and resist damage caused by stresses. It can regulate processes such as plant aging, leaf formation, and fruit ripening [55,56]. The colonization of *S. indica* into Tartary buckwheat resulted in an increase in the content of soluble sugar in the leaves (Figure 6A), enhanced the stress resistance and promoted the growth of Tartary buckwheat plants. Meanwhile, previous studies have shown that soluble sugars can enhance the rooting ability of plants [57], which is consistent with the results of our research. The colonization of *S. indica* in Tartary buckwheat increased the dry weight of roots (Table 2). Soluble proteins are important nutrients and osmoregulatory substances, and their accumulation can improve the water holding capacity of cells, protect the vital substances and biofilms of cells, and is often used as one of the indicators for screening resistance [58]. Plants under stress can participate in balancing the osmotic potential of pants and in reducing the harm caused by stress to the plant. Both low temperature and poor water quality may both stress the production of soluble proteins. The colonization of *S. indica* in Tartary buckwheat extremely significantly increased the content of soluble protein in the leaves (Figure 6B), enhancing the stress resistance of the Tartary buckwheat.

Numerous studies have been carried out on the resistance of *S. indica*. The *S. indica* colonization of *Gerbera jamesonii* enhances its salt tolerance by reducing oxidative damage and increasing the activity of antioxidant enzymes [20]. The colonization of onion by *S. indica* can elevate its antioxidant enzyme activity and induce the expression of defense-related genes to resist foliar blight disease [24]. The results of this study indicate that *S. indica* enhance the drought resistance of Tartary buckwheat plants in terms of overall phenotype. Under drought and normal conditions, the stomatal analysis of *S. indica**colonized and non-colonized Tartary buckwheat leaves revealed that *S. indica* increased the number of stomata per unit area of Tartary buckwheat leaves (Figure 7F), which is beneficial for buckwheat to adjust stomatal opening and closing to cope with drought conditions. 

It has been previously reported that under stress conditions, *S. indica* can induce increased antioxidant enzyme activity in host plants [19,59]. Due to the overall resistance of *S. indica* to drought (Figure 7A), we measured the antioxidant enzyme activity in the leaves of *S. indica*-colonized and non-colonized Tartary buckwheat plants under drought and normal conditions. In summary, the results of this study indicate that under drought stress, the colonization of *S. indica* in Tartary buckwheat significantly increased the GSH-Px, SOD, CAT and POD activities of Tartary buckwheat leaves (Figure 9). POD activity can used as marker and predictor of rooting performance, and the up-regulation of POD activity in *S. indica*-colonized Tartary buckwheat roots indicated that the fungus colonization enhanced the rooting ability [16]. Quantitative analysis of genes related to the induction of antioxidant enzyme synthesis revealed that colonization by *S. indica* increased gene expression and promoted antioxidant enzyme synthesis (Figure 9E).

Antioxidant enzymes can help Tartary buckwheat resist oxidative damage, so we measured several common oxidative markers. MDA is one of the indicators for measuring cytoplasmic membrane peroxidation [60]. In presence of drought stress, *S. indica*-colonized Tartary buckwheat reduced the MDA content of Tartary buckwheat leaves (Figure 8C). This indicates that it reduced the degree of damage to the cell membrane and protected the cell membrane from oxidative injury. Under drought stress, *S. indica* may protect plants by enhancing their antioxidant and free radical scavenging abilities [61]. We observed that the H_2_O_2_ content in *S. indica*-colonized Tartary buckwheat was significantly lower than that in the non-colonized control group (Figure 8B). The removal of H_2_O_2_ can prevent lipid peroxidation, which can explain why colonization by *S. indica* can reduce the MDA content of Tartary buckwheat. As we know, CAT is an H_2_O_2_ scavenger capable of catalyzing the decomposition of H_2_O_2_. Colonization by *S. indica* resulted in an extremely significant increase in CAT activity (Figure 9B) and elevated plant antioxidant capacity. Under drought stress, the Pro content of *S. indica*-colonized Tartary buckwheat extremely significantly increased (Figure 8D). As an osmotic regulator, Pro can prevent cellular dehydration and provide normal plant growth under drought conditions [62]. These results indicated that colonization by *S. indica* improved the drought resistance of Tartary buckwheat.

The colonization of Tartary buckwheat by *S. indica* reduced the content of MDA and H_2_O_2_ in the leaves of Tartary buckwheat, which in turn reduced oxidative damage (Figure 8). To prevent tarnished Tartary buckwheat from being inactivated due to drought and dehydration, the content of the osmotically regulating substance Pro has been increased (Figure 8). In addition, *S. indica* was found to increase the activity of antioxidant enzyme and the expression level of related genes, improved the rooting ability and antioxidant capacity of Tartary buckwheat plants, thereby promoting the growth of Tartary buckwheat (Figure 9).

## 5. Conclusions

Based on the results obtained in this study, we propose a preliminary model to recapitulate the growth of Tartary buckwheat colonized by *S. indica* (Figure 10). Firstly, *S. indica* can colonize Tartary buckwheat roots. Secondly, it promotes the accumulation of chlorophyll in the leaves of Tartary buckwheat, which in turn promotes its photosynthesis and increases its biomass. Thirdly, the Tartary buckwheat colonized by *S. indica* also regulates the plant hormone metabolic pathway, stimulates the synthesis of IAA and the ethylene precursor ACC, and promotes the growth of Tartary buckwheat. In addition, the colonization by *S. indica* promotes the accumulation of soluble sugars, soluble proteins, total flavonoids, and total phenols in Tartary buckwheat, thereby improving its stress resistance. At the same time, the *S. indica* colonization of Tartary buckwheat reduces the content of MDA and H_2_O_2_ in leaves, and increases the content of Pro and the activity of antioxidant enzymes, increasing the systemic resistance of Tartary buckwheat and ultimately promoting the growth of the entire plant. 

## Figures and Tables

**Figure 1 jof-09-01114-f001:**
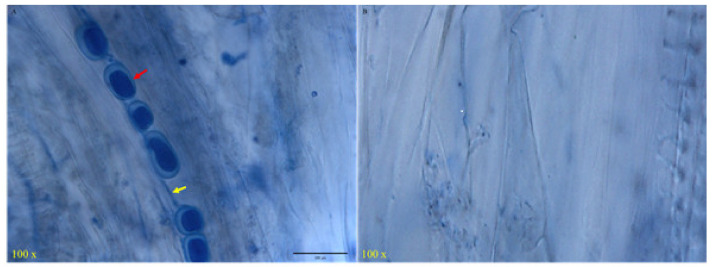
*S. indica*-colonized (**A**) and non-colonized (**B**) detection result. The red arrow represents the spores of *S. indica*; the yellow arrow represents the mycelium of *S. indica*.

**Figure 2 jof-09-01114-f002:**
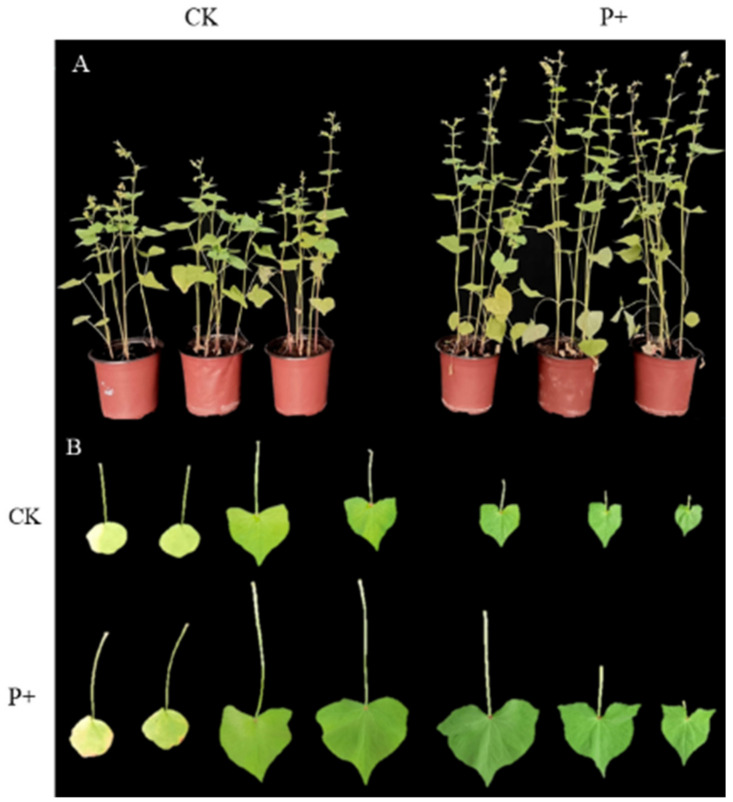
Tartary buckwheat inoculated with *S. indica*. (**A**) Control (CK) and *S. indica*-colonized (P+) Tartary buckwheat plants; (**B**) comparison of the first to seventh leaves of CK and P+ Tartary buckwheat plants.

**Figure 3 jof-09-01114-f003:**
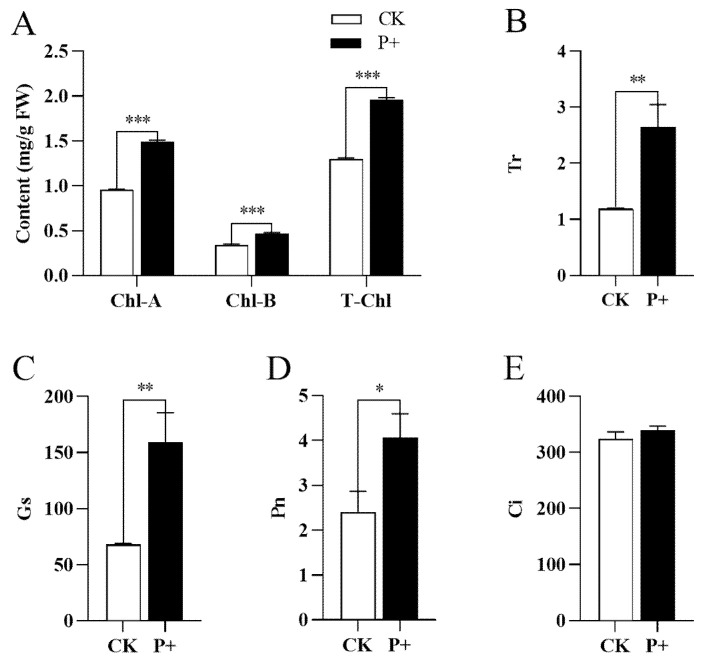
(**A**) Chlorophyll content of *S. indica*-colonized (P+) and non-colonized (CK) Tartary buckwheat plants; Cellular respiration parameters related to colonized (P+) and non-colonized (CK) of *S. indica*: (**B**) Transpiration rate (Tr); (**C**) Stomatal conductance (Gs); (**D**) Net photosynthetic rate (Pn); (**E**) Intercellular CO_2_ concentration (Ci). Values are presented as mean ± SD (standard deviation) of three replicates. * *p* < 0.05; ** *p* < 0.01; *** *p* < 0.001.

**Figure 4 jof-09-01114-f004:**
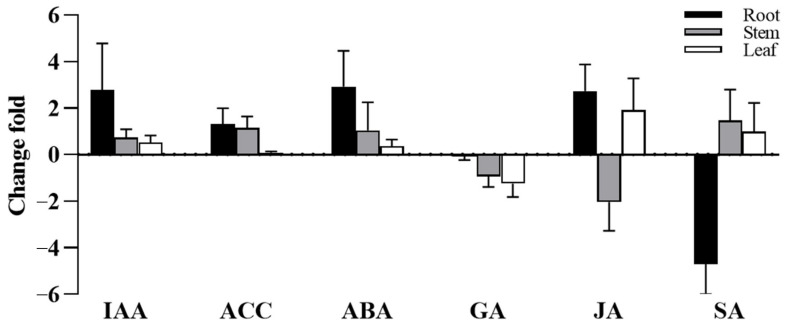
The change fold of plant hormone in root, stem, and leaf of Tartary buckwheat colonized and non-colonized by *S. indica* (The change fold refers to the multiple of hormone content in various parts of *S. indica*-colonized Tartary buckwheat compared to that in non-colonized Tartary buckwheat). Values are presented as mean ± SD (standard deviation) of three replicates.

**Figure 5 jof-09-01114-f005:**
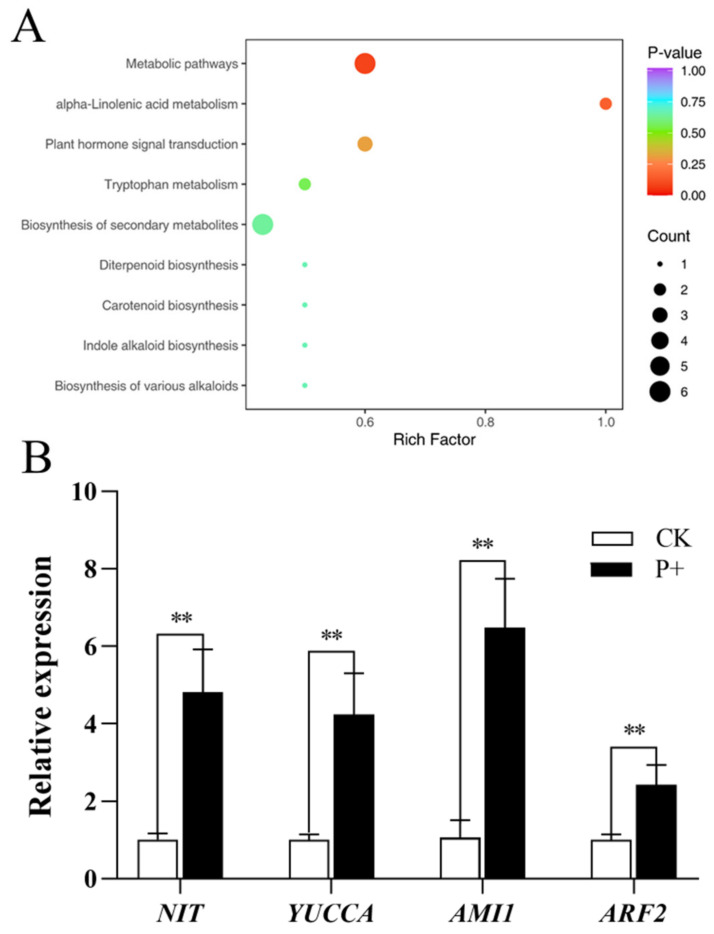
(**A**) Enrichment map of differential metabolite KEGG. The horizontal axis represents the Rich factor corresponding to each pathway, the vertical axis represents the pathway name, and the color of the point is *p*-Value. The redder points are the more significant the enrichment. The size of the point represents the number of enriched differential metabolites; (**B**) The expression levels of genes related to the auxin synthesis pathways in *S. indica*-colonized (P+) and non-colonized (CK) Tartary buckwheat. Values are presented as mean ± SD (standard deviation) of three replicates. ** *p* < 0.01.

**Figure 6 jof-09-01114-f006:**
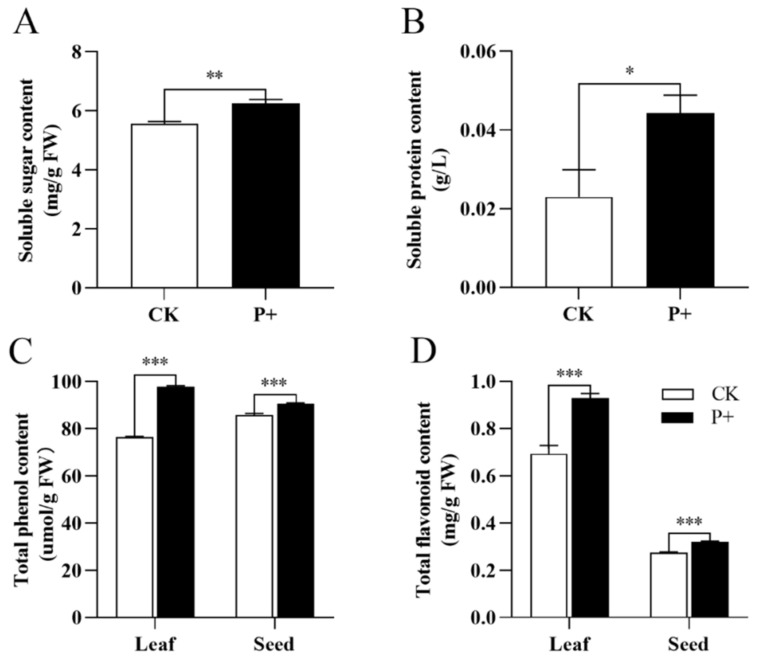
(**A**) Soluble sugar and (**B**) soluble protein content of *S. indica*-colonized (P+) and non-colonized (CK) Tartary buckwheat. The content of total phenols (**C**) and total flavonoids (**D**) in leaves and seeds of *S. indica*-colonized (P+) and non-colonized (CK) Tartary buckwheat. Values are presented as mean ± SD (standard deviation) of three replicates. * *p* < 0.05; ** *p* < 0.01; *** *p* < 0.001.

**Figure 7 jof-09-01114-f007:**
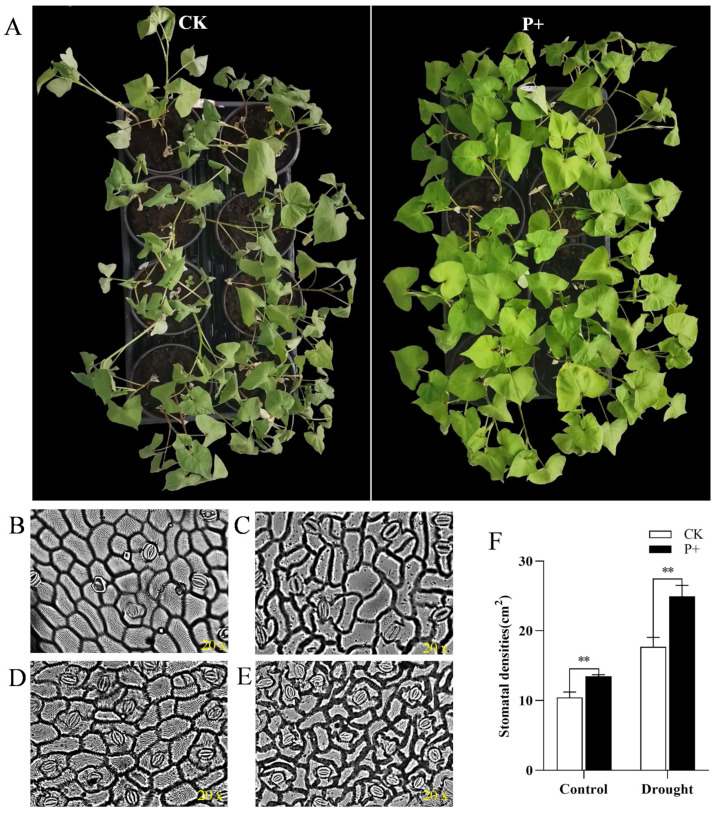
(**A**) Comparison of the overall phenotypes of *S. indica*-colonized (P+) and non-colonized (CK) Tartary buckwheat plants under drought stress. Under different conditions, the stomata of Tartary buckwheat leaves are: (**B**) without drought and *S. indica*-colonized; (**C**) no drought with *S. indica*-colonized; (**D**) drought with *S. indica* non-colonized; (**E**) drought with *S. indica*-colonized. (**F**) The stomatal densities of *S. indica*-colonized (P+) and non-colonized (CK) buckwheat leaves. ** *p* < 0.01.

**Figure 8 jof-09-01114-f008:**
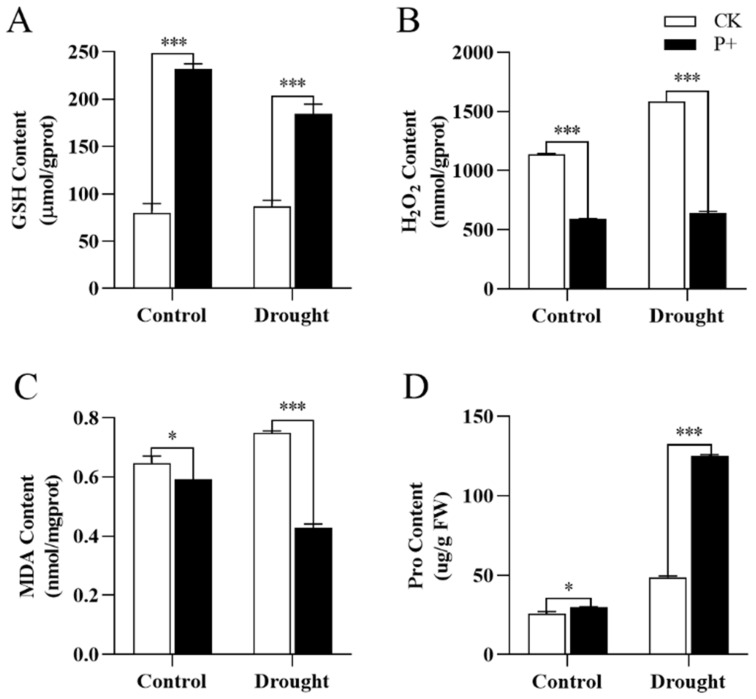
Under drought stress (drought) and normal conditions (control), the content of stress related factors in the leaves of *S. indica*-colonized (P+) and non-colonized (CK) Tartary buckwheat. (**A**) GSH content; (**B**) H_2_O_2_ content; (**C**) MDA content; (**D**) Pro content. Values are presented as mean ± SD (standard deviation) of three replicates. * *p* < 0.05; *** *p* < 0.001.

**Figure 9 jof-09-01114-f009:**
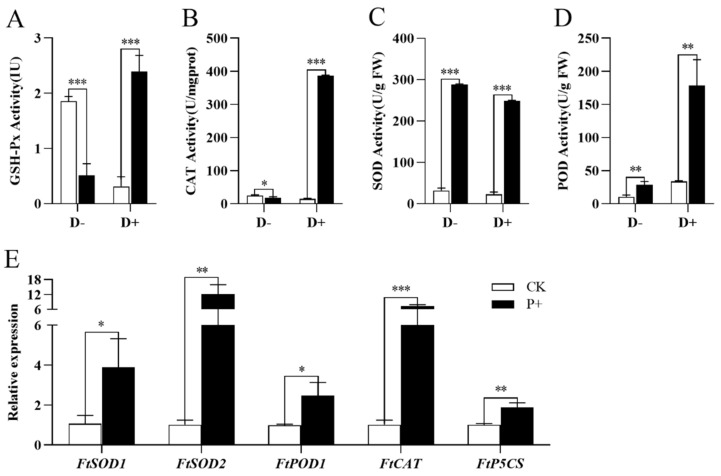
Under drought stress (D+) and normal conditions (D−), the activity of stress-related enzymes in the leaves of *S. indica*-colonized and non-colonized Tartary buckwheat. (**A**) GSH-Px activity; (**B**) CAT activity; (**C**) SOD activity; (**D**) POD activity. (**E**) Under normal conditions, the expression levels of antioxidant-related genes in the leaves of *S. indica*-colonized (P+) and non-colonized (CK) Tartary buckwheat. Values are presented as mean ± SD (standard deviation) of three replicates. * *p* < 0.05; ** *p* < 0.01; *** *p* < 0.001.

**Figure 10 jof-09-01114-f010:**
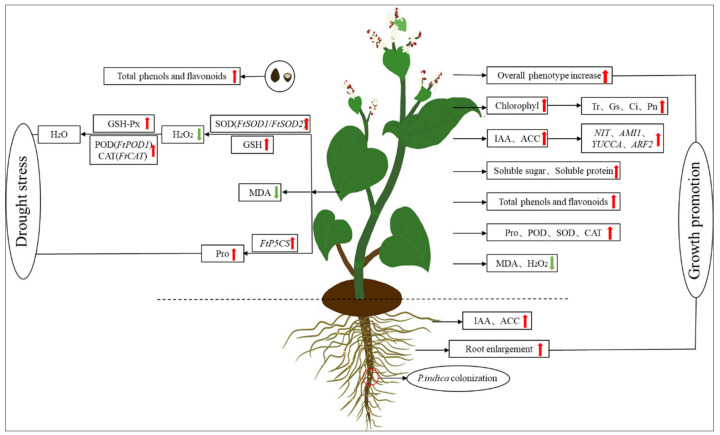
Summary diagram for the growth promoting effects of *S. indica* on Tartary buckwheat. Red and green arrows represent parameters induced and suppressed by *S. indica* colonization, respectively.

**Table 1 jof-09-01114-t001:** Primer sequences of Tartary buckwheat-related genes.

Gene	Sequence	Primers
*FtCACS*	F′	AAGACAGTCAGTTTCGTGCCACCTGA
	R′	TCCATGCGTGTTCTACCCAACTCCTT
*FtAMI1*	F′	CACGGCTTCGGCAGTTCA
	R′	TGGGTTTCTGGGTGTCCC
*FtYUCCA10*	F′	TTTCCCGATCACTTTCCG
	R′	GCCGTCGTTGTCTTAACTCT
*FtNIT*	F′	CGGCCAAATCGACACTCC
	R′	CCACCTTTCGCTGCTTCC
*FtARF2*	F′	AGACTTGTGGCTGGTGACGCT
	R′	GCTAGATATGACTGACGAGGGAACT
*FtCAT*	F′	GGAGGAGCAAACCACAGT
	R′	TTCAAGACCATCCGACCC
*FtSOD1*	F′	AAGCCGCCATTCTCACTA
	R′	ACAACGCCTTCAACATCG
*FtSOD2*	F′	TGAAGGCTGTAGTTGTTCTC
	R′	ACCCATTGGTGGTGTCCC
*FtPOD1*	F′	AACTGTGCTCCGATTATGC
	R′	AGCTCCTCCTGGAACCTC
*FtP5CS1*	F′	CAAGGATGTCAAGCGTAT
	R′	CAGCACCTGAACTCACTAAA

**Table 2 jof-09-01114-t002:** Effects of *S. indica* on plant height, leaf length, leaf width, petiole length, aboveground part fresh and dry weight, root fresh and dry weight, and hundred-grain weight of Tartary buckwheat plants. Values marked with “*” in each row represent that the difference between samples were significant (Student’s *t*-test, ** *p* < 0.01; *** *p* < 0.001; n = 4).

Growth Parameters	Control	*S. indica*
Plant height (cm)	26.17 ± 3.47	47.59 ± 3.60 ***
Leaf length (cm)	2.61 ± 0.05	4.13 ± 0.10 ***
Leaf width (cm)	2.78 ± 0.10	4.42 ± 0.25 ***
Petiole length (cm)	2.78 ± 0.09	5.22 ± 0.21 ***
Aboveground part fresh weight (g)	2.08 ± 0.09	3.06 ± 0.29 **
Aboveground part dry weight (g)	0.42 ± 0.01	0.60 ± 0.01 ***
Root fresh weight (g)	0.29 ± 0.02	0.57 ± 0.03 ***
Root dry weight (g)	0.05	0.09 ± 0.01 ***
Hundred-grain weight (g)	1.61 ± 0.04	1.77 ± 0.03 **

## Data Availability

Data are contained within the article.

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
