# Peer review of "Serendipita indica Promotes the Growth of Tartary Buckwheat by Stimulating Hormone Synthesis, Metabolite Production, and Increasing Systemic Resistance"

_jof, 2023, doi:10.3390/jof9111114_

Round 1

Reviewer 1 Report

Comments and Suggestions for Authors

The work reported is interesting. However, there are some points that the authors need to address for publication. The points are given below:

Abstract: it is a well written, but it is necessary to add into the objective, not only growth, but also hormones, metabolites…  

Introduction: the last sentence should be removed to methods for what the authors compare, lines 66-74

Materials and methods

-Make the word “ddH2O” clear

-Line 88: One group, how many plants for one group.

-Section 2.3, please add how to measure these parameters

-Section 2.4-2.10, please cite the references

-Table 1, avoid characteristic “A” in one line

Results

-line 221, n=4, please describe in methods

-Both Chlorophyll A or a; B or b can use, but please uniform in manuscript

-Figure 4, several parameters as ACC, ABA, GA, JA and SA need to describe in methods

Discussion

-This is well design, but it is need to indicate your results; For example, in lines 451-452, which table or Figure belong to

Conclusion

Please delete lines 523-529 as it background, hypothesis, result

Reference

Please check the character “J” in all references

Comments on the Quality of English Language

Minor editing of English language required

Author Response

Dear reviewer,

Thank you very much indeed for your excellent comments on our manuscript. Those comments are valuable and very helpful. We have read through comments carefully and have made corrections. Based on the instructions provided in your letter, we uploaded the file of the revised manuscript. Revisions in the text are shown using red for additions.

Comments 1: Abstract: it is a well written, but it is necessary to add into the objective, not only growth, but also hormones, metabolites

Response 1: I have added a description in the abstract of the impact of P. indica colonization on the changes in total phenols and flavonoids in the leaves and seeds of Tartary buckwheat.

Comments 2: Introduction: the last sentence should be removed to methods for what the authors compare, lines 66-74

Response 2: I have deleted the description of the method in the last paragraph of the introduction.

Comments 3: Materials and methods: Make the word “ddH2O” clear

Response 3: I have changed the ddH2O in the method to double distributed water.

Comments 4: Materials and methods: Line 88: One group, how many plants for one group.

Response 4: I have added relevant descriptions: the P. indica group and the control group each have eight pots, with four plants per pot.

Comments 5: Materials and methods: Section 2.3, please add how to measure these parameters

Response 5: I have added a specific description of the measurement method.

Comments 6: Materials and methods: Section 2.4-2.10, please cite the references

Response 6: I have added references to all experimental methods.

Comments 7: Materials and methods: Table 1, avoid characteristic “A” in one line

Response 7: I have made modifications to Table 1.

Comments 8: Results: line 221, n=4, please describe in methods

Response 8: I have added relevant descriptions to the method.

Comments 9: Results: Both Chlorophyll A or a; B or b can use, but please uniform in manuscript

Response 9: I have uniformed the relevant descriptions in the manuscript.

Comments 10: Results: Figure 4, several parameters as ACC, ABA, GA, JA and SA need to describe in methods

Response 10: I have added relevant descriptions in the method section.

 Comments 11: Discussion: This is well design, but it is need to indicate your results; For example, in lines 451-452, which table or Figure belong to

Response 11: I have added my results in the discussion of the relevant descriptions.

 Comments 12: Conclusion: Please delete lines 523-529 as it background, hypothesis, result

Response 12: I have removed the section on background description from the conclusion, which currently contains a brief description of Figure 10.

Comments 13: Reference: Please check the character “J” in all references

Response 13: I have made modifications to the format of all references.

Reviewer 2 Report

Comments and Suggestions for Authors

 The article “Piriformospora indica promotes the growth of tartar buckwheat by stimulating hormone synthesis, metabolite production and increasing systemic resistance” by Zheng et al., deals with the effects of the endomycorrhizal fungus P. indica in the roots of Fagopyrum tataricum. In recent decades P. indica has been indicated as a fungus capable of promoting plant growth and, in this case, it has been tested to evaluate its effects on the host. Although the anatomical and physiological investigations highlight data of particular interest, numerous methodological errors are found in this work.

First, endophytic fungus inoculations should be performed on plants grown on sterile and non-sterile substrates. In this work the treatments are carried out only on non-sterile substrates, so much so that photo 1b shows a fungal mycelium in the root of a non-inoculated (non-colonised) plant. This invalidates all resulting data. Furthermore, the introduction does not contain the bibliography relating to the studies on the fungal endophytes of F. tataricum, which had already begun previously. Furthermore, it is not reported that this work investigates the possible colonization of P. indica in F. tataricum plants which, on the contrary, is taken for granted. Finally, the introduction is confusing, does not provide a logical sense of the development of the work and includes completely useless information, while it does not delve into useful data.

Author Response

Dear reviewer,

Thank you very much indeed for your excellent comments on our manuscript. Those comments are valuable and very helpful. We have read through comments carefully and have made corrections. Based on the instructions provided in your letter, we uploaded the file of the revised manuscript. Revisions in the text are shown using red for additions.

Comments 1: First, endophytic fungus inoculations should be performed on plants grown on sterile and non-sterile substrates. In this work the treatments are carried out only on non-sterile substrates, so much so that photo 1b shows a fungal mycelium in the root of a non-inoculated (non-colonised) plant. This invalidates all resulting data. 

Response 1: As for the results of the colonization of Tartary buckwheat by P. indica in Fig. 1A, I did this under sterile conditions and planted the sterilized seeds into the soil after autoclaving. I am very sorry for the misunderstanding caused to you because I did not write it in the method. I did not co-culture the Tartary Buckwheat plants tested in Fig. 1B with P. indica, and the mycelia like things appeared, which may be due to the problems in my dyeing process. In addition, the obvious chlamydospore of P. indica in Fig. 1A, which more clearly indicates that P. indica can colonize in Tartary buckwheat roots, laying a foundation for subsequent experiments.

 Comments 2: Furthermore, the introduction does not contain the bibliography relating to the studies on the fungal endophytes of F. tataricum, which had already begun previously. 

Response 2: Thank you for your suggestion. I have included the relevant research between Tartary buckwheat and endophytic fungi in the introduction section.

Comments 3: Furthermore, it is not reported that this work investigates the possible colonization of P. indica in F. tataricum plants which, on the contrary, is taken for granted.

Response 3: Of course. P. indica are fungi that can be widely customized. Our original intention was not to see if P. indica could colonize on Tartary buckwheat, but to solve the problems encountered during the growth of Tartary buckwheat. The growth of Tartary buckwheat has a series of problems, such as poor soil quality and low seed activity in the planting area. We hope that P. indica can solve these problems. Of course, other bacteria also have certain effects on the growth of Tartary buckwheat, but relatively speaking, P. indica can solve more comprehensive problems.

Comments 4: Finally, the introduction is confusing, does not provide a logical sense of the development of the work and includes completely useless information, while it does not delve into useful data.

Response 4: Thank you for your advice. I have revised the relevant logic in the introduction.

Round 2

Reviewer 2 Report

Comments and Suggestions for Authors

The Article: Piriformospora indica promotes the growth of Tartary buckwheat by stimulating hormone synthesis, metabolite production, and increasing systemic resistance, by Zheng et al. 

The following changes are recommended.

The fungal specie Piriformospora indica has been revised as Serendipida indica (see bibliographic reference in the text) and appropriate changes should be made in the text.

Line 89: report the cited bibliographical reference;

Lines 102-104, 116-117 and 122: Sentences should be written more smoothly (not "Use" or "Dry").

Furthermore, a file with the changes relating to the abstract, introduction and conclusions is attached.

Author Response

Dear reviewer,

Thank you very much indeed for your excellent comments on our manuscript. Those comments are valuable and very helpful. We have read through comments carefully and have made corrections. Based on the instructions provided in your letter, we uploaded the file of the revised manuscript. Revisions in the text are shown using red for additions.

Comment 1: The fungal specie Piriformospora indica has been revised as Serendipida indica (see bibliographic reference in the text) and appropriate changes should be made in the text.

Response 1: I have changed all the names of Serendipida indica in the text.

Comment 2: Line 89: report the cited bibliographical reference;

Response 2: The specific operation of this method was improved and taught by the laboratory senior based on relevant principles, without specific references.

Comment 3: Lines 102-104, 116-117 and 122: Sentences should be written more smoothly (not "Use" or "Dry").

Response 3: I have made modifications to the description in the article.

Comment 4: Furthermore, a file with the changes relating to the abstract, introduction and conclusions is attached.

Response 4: Thank you for your suggestion. I have corrected the relevant content.